# Critical Reflection on Inequality and Precariousness in Contemporary Society: Reconstructing Connections from Social Intervention

**María del Carmen Sánchez-Miranda** [1], **José-Luis Anta-Félez** [1], **Romina Grana** [2] and **Rubén Gregorio Pérez-García** [3,*]

[1] Department of Anthropology, Geography and History, Faculty of Humanities and Communication Sciences, University of Jaén, Jaén, 23071, Spain; mmiranda@ujaen.es (M.d.C.S.-M.); jlanta@ujaen.es (J.-L.A.-F.)

[2] Research Centre of the Faculty of Philosophy and Humanities, University of Córdoba, Córdoba 5000, Argentina; romina.grana@unc.edu.ar

[3] Department of Psychology, Faculty of Social Work, University of Jaén, Jaén, 23071, Spain

[*] Correspondence: rgperez@ujaen.es; Tel.: +34 953211768

**Abstract:** This article, oriented towards good practices in social action, is nourished by reflection on the marked inequalities and precariousness that characterise contemporary reality, focusing on how these dynamics undermine the equitable distribution of opportunities that are fundamental for a dignified existence. From this perspective, we seek to reflect on the dimensions of precariousness and inequality, allowing for the re-evaluation of interventions in different environments and critically addressing the possibility of new modalities of action. The proposed approach implies an ethical commitment to overcoming the dilemma between adhering to the logic of the market and adopting a more humane and supportive stance, challenging conventions and promoting socially responsible practices. In short, the aim is to analyse intervention, enabling the (re)understanding of inequality in the social context and questioning the social structures that generate suffering, with a view to reconstructing the essential concepts of solidarity and community in contemporary society, in a critical understanding of socio-educational action in society.

**Keywords:** social intervention; inequalities; precariousness; labour/educational spheres; political context

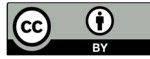

## 1. Introduction

This work arises from a deep reflection that is centred around the field of social action, whose general objective is to reflect on various forms of intervention in the complex labour and educational context that characterises our current reality. Reflecting on the world we live in, its marked inequality and precariousness stand out as prominent characteristics [1]. Inequality manifests itself as the first of these characteristics, while precariousness constitutes the second. A priori and disregarding moral judgements, one could argue that neither should be a cause for concern. In this sense, the lack of uniformity among people would not necessarily generate concern, and the ephemeral nature of our philosophical existence would suggest that the precariousness of the world is a reflection of our temporality.

However, the apparent reassurance of these characteristics fades when one examines an underlying concern: the inequality and precarity of the world becomes relevant as it hinders the equitable distribution of essential employment and educational opportunities for every person to lead an existence with a minimum of dignity. Precarity limits our abilities to assert rights and fulfil obligations, and inequality takes on meaning when

intertwined with precarity, underscoring the urgency of addressing these issues in a holistic manner.

Reflections on these dimensions constitute unavoidable aspects for those of us who, from our professions and social tasks, are concerned with the conditions that make possible the emergence of the subject in certain contexts from ontological ontologies of being and contexts from a critical stance [2]. It is for this reason that throughout the following work, rather than definitive answers on how the social question is presented, we offer some reflections on how it has developed in a socio-historical perspective that allows us to build bridges of interpretation on our contemporary condition.

## 2. Some Historical Enclaves for Thinking about Inequality and Precariousness

If we ask ourselves how we have arrived at the current situation, it is imperative to reflect on the events and processes that have led to the present configuration. Some precedents that allow us to address this question can be found, in the context of the 19th century, in thinkers such as Carl Schmitt [3], David Ricardo [4] and Karl Marx [5], who dedicated their reflections to the analysis of the economy, politics and society of their time from a materialist view of historical development and a dialectical perspective of social transformation. Having examined the conditions of their time, they identified an extremely difficult and complex reality of existence. Obtaining employment and securing a living wage was a challenge, as everyday life was characterised by a lack of labour regulation, the absence of standardised working hours and the lack of fundamental rights, such as the right to holidays. This complexity of living conditions led these leading 19th century economists to consider the reality of their time to be, simply put, frightening.

It is important to note that the answer to the question of how we have arrived at the current situation is not encapsulated in a single explanation. The diversity of historical and social phenomena suggests that multiple forces and factors contributed to the development of contemporary reality. In our present, unlike the tragic reality of the 19th century, we have managed, to some extent, to mitigate the severity of living conditions through the relativisation of reality and the construction of narratives that disguise adversities [6].

Despite the divergences between the circumstances of the past and our present, it is undeniable that the situations of the 19th century were genuinely tragic. The non-existence of the notion of childhood [7] (which had a strong impact on the conception of the pupil that developed in schools) meant, for example, that boys and girls entered the world of work without any regulation, which led to extremely precarious working conditions for them. In this sense, the existence of the idea of childhood was called into question and the transition to adulthood was rushed, without consideration for children's well-being. In this context, the lack of clear regulations regarding child labour perpetuated situations of exploitation and the lack of fundamental rights [8] as well as having repercussions in forging a conception of the "adult" pupil who was asked to do things that, as a child, he or she could not do.

Two fundamental ideas characterised the nineteenth century and generated continuities throughout the twentieth century [9]: first, it highlights the magnitude of inequalities and disparities between people, a reality that can generate deep consternation when considering large-scale figures. For example, approximately 1% of the world's population owns around 82% of the world's land [10]. This is even more shocking when we realise that, for example, when we enter our homes, the entirety, with the exception of the shower tray, belongs to other individuals, and the family must inhabit that shower tray. This image exemplifies the tragedy of 1% of the population owning the totality of resources, from material goods to control of media and businesses.

This fraction of the population not only owns property but also substantially influences the design and functioning of the social model in which we live [11]. A fundamental aspect of this lies in the fact that we inhabit an environment that has been meticulously structured by a group of people who lack an empathetic connection with the vast majority

of the population. This elite, being the driving force behind the design of the system, determines the rules and dynamics that govern our daily lives. Secondly, another tragedy manifests itself in the reflection on the personal consequences of this unequal reality. If an individual belonging to this small percentage were to decide to acquire the entirety of a territory, the impact would be devastating. Not only would this act displace entire communities, but it would also nullify the possibility of recreating the intrinsic value of the territory in a new place. This underlines the inherent injustice of a system in which a minority acquires the power to decide the lives and destiny of the majority without considering the consequences and the significant value that different regions hold for their inhabitants.

Consequently, we find ourselves immersed in a context of marked inequality, which not only manifests itself at the macro-structural level but also permeates more intimate aspects of our society [12]. Disparities are evident in multiple dimensions, from those who have access to education and those who do not to differences in the ownership of assets such as vehicles or the variability in working conditions, whether in terms of full-time or part-time employment. It also extends to people's ability to meet their monthly economic needs and to small variations in culture, such as the number of books, records or theatre visits they are able to make.

It is imperative to recognise that these small, seemingly trivial differences shape significant divergences in the quality of life and the perceived meaning of existence. It is crucial to realise that our everyday environment, and therefore the environment in which we intervene, is intrinsically marked by these inequalities, which in turn are rooted in what we call "fields of action". These fields represent territories with which we identify ourselves and where we develop our interactions [13]. Contrary to superficial perception, our disputes and challenges are not centred on public figures or celebrities, but rather, they are rooted in our closest environments: family, neighbourhood, classmates and other close circles. In this sense, it is relevant to highlight that our competitions and confrontations unfold within these fields of action where inequalities are not only perpetuated but even become spaces conducive to other ends, whether to claim rights, ask for help or generate new social dynamics [14].

Moreover, these interactions do not occur in an abstract vacuum, but develop over time, guided by agendas that delineate with whom we engage and what the goals of our social exchanges are [15]. In this way, an intricate web of relationships and conflicts is woven that defines our everyday experiences. Critical theorists have conceptualised the phenomenon as a "field of contestation" [16], noting that these are spaces of confrontation over truth. In this context, truth becomes the object of dispute, where seemingly trivial everyday interactions are transformed into arenas of struggle over the perception of reality. These arenas of dispute, however, are not limited only to discussions about abstract truths; rather, they become concrete in everyday aspects, such as the acquisition of material goods. An illustrative example would be criticism of a neighbour who has acquired a luxury car. This criticism is not reduced to a disagreement about the truth but reveals a conflict over certain pre-established values and criteria. By revealing the supposed truth behind the external appearance, it seeks to highlight discrepancies and sometimes to mask feelings of envy or rivalry.

The act of removing the mask, in this context, involves unravelling the underlying reality of these disputes. However, a further complexity arises when it is recognised that, to a large extent, all of us, without exception, have been emptied of content. This emptying implies that, in stripping away the truth, an authentic essence is not revealed; rather, a void is found. This phenomenon constitutes one of the most serious problems in the fields of dispute, since struggles for truth are fought on a terrain devoid of real substance. This hollowing out compares to the superficial representation that many people project on social media, such as Instagram. The lives we present publicly often lack depth and real content. Similarly, the truths we pursue and the criticisms we make are framed by superficial masks that hide the lack of authenticity and substance behind our daily interactions.

This phenomenon adds complexity to the task of expressing and receiving feedback, as ultimately our lives and the truths we seek are essentially devoid of genuine content.

### 3. (Re)Considering Inequality and Precariousness: New Perspectives on the Labour Market

#### 3.1. A Gender Issue

In this context, inequality manifests itself both on large numerical scales and in everyday interactions that shape a more subtle and closer inequality [17]. In turn, another significant theme that deserves attention is precarity, a social reality intrinsically linked to the world of work, but one that encompasses broader dimensions than simply the realm of work. Precarity, in essence, implies the ability to establish a meaningful life, from the ability to acquire material goods to the financial freedom to engage in various cultural activities, buy a house or pursue personal dreams [18]. Although the centrality of precarity lies in the realm of work, it extends to broader issues of establishing a life. In the current era, where artificial intelligence dominates our social and work interactions, algorithms play a crucial role in determining fundamental aspects of our lives. This is reflected, for example, in the ability to obtain a mortgage, where artificial intelligence algorithms can influence the bank's decision based on a variety of criteria, including gender, occupation and place of residence, among others. This dialogue between machines and us adds a variable to the dynamics of precarity; as algorithms become arbiters of our lives, situations such as applying for a mortgage may depend on criteria that may not objectively reflect the applicant's reality. For example, the current tendency of some algorithms to deny mortgages to women [19] illustrates how these tools can perpetuate gender biases and inequalities.

It is crucial to understand that, even if we have not directly experienced this situation, the growing role of artificial intelligence in everyday decision-making raises important questions about fairness and justice in an increasingly automated world [20]. Gender-based discrimination, such as the one mentioned above, lacks an objective reason from a fair and just perspective. However, this inequality is rooted in social constructions and power structures, particularly situated in a machista and heteropatriarchal context. Gender misconceptions and stereotypes are reflected and perpetuated by algorithms designed in a social environment that favours certain biases.

If we pay attention to the level at which language reproduces these biases, the case of online product sales platforms such as Amazon, whose search engine is able to detect gender (and this in binary terms) because the database on which the software relies contains a large percentage of data referring to males, clearly emerges.

Going even further, being a woman, according to these stereotypes and many others that are perpetuated in the social space, is associated with assumptions such as the possibility of becoming pregnant and, therefore, receiving fewer employment benefits or being considered less productive because she is taking care of her children. In addition, there is a perception that women are more malleable in salary negotiations, which could lead to lower pay and, consequently, greater difficulties in meeting financial commitments such as mortgage payments. And even for early years education, there is a stereotype that it is women who are better suited to the role of teacher, as this role is associated with their possible maternal nature.

It is crucial to highlight that these prejudices have no basis in reality but are social constructs that influence the configuration of algorithms and automated decision-making systems. In this sense, gender discrimination is not only reflected in the denial of mortgages but also in the differentiated obtaining of other financial products, which is further evidence of the complex web of discriminations and stereotypes managed by algorithms and technologies, which, although perceived as objective, are inherently biased by the values and social constructions that underlie their development. Moral violence is reproduced with automatism and invisibility for a long time after its establishment, which

allows for inertia and naturalisation when it is exercised, as part of behaviours considered normal and banal, and its rootedness in moral, religious and family values, which allows its continued justification. This mechanism, or rather device, does nothing more than generate categories for looking at the social world and perpetuating strategies of control; in Segato's words, it is about managing, time and again, "vows of subordination of the minoritised in the status order and the permanent concealment of the instaurator's act" [21].

### 3.2. A Question of Machines

This aspect becomes particularly relevant when we consider that, for the last 150 years or so, humanity has been engaged in a constant struggle with machines [22]. These machines, whether highly complex like the one that makes our current interconnectedness possible or simple devices in charge of routine tasks, play a fundamental role in the production of practically all the goods we own. The presence of machines spans a range of domains, from highly sophisticated technologies that facilitate virtual interaction to those responsible for more basic tasks, such as stapling documents. What is significant, however, is that these machines not only occupy a physical space in production but have also substantially influenced work structures.

In the past, work structures were marked by interdependencies, with each person playing a crucial role in the production chain [23]. The manager had to lead efficiently to ensure that each employee performed his or her task accurately, as individual actions had a direct impact on the final result. However, these labour interdependencies have been progressively replaced by machines, which has led to significant changes in labour dynamics. A telling example of this change is the modern-day virtual classroom [24]. Although the convenience of this format is undeniable, it is essential to reflect on the implications in terms of employment and appropriation on the part of students. Regarding the former, the transition to this model has left out numerous workers, such as the cleaning staff at the university or those in charge of maintaining and distributing the necessary devices. This apparent ease of virtual connection has led to an increase in precarious work quotas, highlighting the need to reflect on the ethical and social dimensions of automation and technology. With regard to the possible appropriation of virtual space by students, what is happening is very relative; digital literacy is needed to take on any kind of education on a digital campus or platform, which is impossible to ensure, for example, in Latin American countries that do not have the devices and connections adequate for the large geographical extensions that separate urban centres from educational institutions in this vast territory.

The maxim in our world that any work that can be carried out by a machine should be assigned to a machine has become a norm we all adhere to. This perspective, while convenient in some respects, such as the use of technologies like Bizum to make payments efficiently, also has social and ethical implications that should not be overlooked [25]. When we incorporate advanced technologies, we benefit from their convenience, but we must be aware that this excludes certain individuals. The automation and digitisation of the payment process may leave out those who do not have access to modern technologies, either due to financial constraints, a lack of familiarity with new technologies or a lack of appropriate devices. This aspect is clearly visible in the age group of older adults, who struggle daily to make friends with bank-based platforms, digital transactions in hospitals, trade unions and social security, among others, which dehumanise everyday practices, undermine dialogue and standardise the type of responses expected.

This trend towards automation also has an impact on the workplace [26]. By making mechanical tasks available to machines, jobs previously held by humans are being eliminated, and even in education, a line of impersonality is being drawn in a relationship—teacher/student—that has always been subjective. While this may increase efficiency and reduce costs, it also creates inequalities, as those without the necessary skills to work with advanced technologies are excluded from the labour market. Moreover, the widespread adoption of technologies can have negative consequences for those who do not have

access to them. For example, the elimination of traditional banking counters may make transactions more difficult for the elderly or those who are unfamiliar with new technologies. In this sense, the constant struggle against machines must be critically analysed. While automation can improve efficiency and simplify certain processes, it is also important to consider its social and economic implications [26]. The adoption of new technologies should not contribute to the creation of inequalities or leave behind disadvantaged segments of society such as those we have mentioned for Latin America. The onus is on us as a society to ensure that the implementation of technologies moves forward in an ethical and equitable manner. Savings from the reduction in cleaning staff do not mean that the money has evaporated; rather, it is directed towards those who have driven and benefit from modern technologies, such as the internet. One of the most significant examples is how artificial intelligence took centre stage at the last World Economic Forum in Davos—accessible through the World Economic Forum (2024)—where the most influential lobbies in today's international economy advocated development in this area of knowledge as a new gateway to economic growth.

Beyond inequality, which has become an almost naturalised feature of our society, precariousness plays a fundamental role [27]. Although it is intrinsically related to labour systems, we are facing a more complex and worrying phenomenon. Its nature goes beyond being simply a consequence of other factors; it has deeper and broader implications in our reality. Precarity, in essence, is connected to uncertainty in work and the ability to maintain a dignified life. As we move towards an increasingly technology-dominated society, precarity refers not only to job insecurity, but also to uncertainty in everyday life, which constitutes an additional layer of vulnerability [28].

These dimensions we have been talking about do not only affect work and education but extend to various aspects of our lives. From the ability to buy a car to the ability to enjoy cultural activities, precariousness permeates every aspect of our existence; machines and artificial intelligence are displacing jobs, creating an even more precarious working environment. In short, it is not simply a question of unstable employment; it is a broader issue that affects our quality of life and highlights structural inequalities in our society [29]. Understanding and addressing precariousness becomes essential to building a more equitable and sustainable future. The internet came with the promise of interconnecting the world in a democratic way. One of these promises was the possibility for all of us to have a voice, to be able to express ourselves regardless of what we had to say. To some extent, this premise has been fulfilled, as evidenced by the presence of YouTubers, even if they speak without having a meaningful message. The second promise was related to the removal of obstacles to fulfilling our desires. Before the internet, social life was full of barriers, whether it was to receiving an education, finding a job or fulfilling our goals. The internet promised to be a platform where we could move freely, express ourselves without restrictions and face few impediments to achieving our goals. However, these promises have come with their own set of problems; as technology has advanced, so have digital divides and new forms of precarious employment. Moreover, the democratisation of internet access does not guarantee equal opportunities or remove structural barriers in society [30].

## 4. Paradigm Shift and New Scenarios: Contributions to Thinking about New Sensibilities

It is important to reflect on how these original promises of the internet have translated into our current reality and how emerging challenges can be addressed to achieve a true democratisation of information and opportunities online. Commercial platforms, born in the 1990s, presented themselves as problem solvers, taking upon themselves the complications of materiality that made everyday life difficult. These "extractive platforms" operate by extracting realities from the marketplace and have transformed the way we deal with everyday challenges [31]. In the realm of food, they solve problems related to the complexity and time needed to obtain food by offering a quick and convenient

solution. In addition, they address the obstacles of getting around cities, especially in large, hard-to-reach places such as those mentioned above. On the streets, transport services have been simplified, as they offer more comfortable and friendly solutions for getting around the city, addressing the problems associated with conventional taxis, such as lack of cleanliness or discomfort. In terms of accommodation, access to temporary housing in unfamiliar places has been transformed; once upon a time, finding a place to stay in a new city was complicated and limited to hotels or guesthouses. Now, platforms allow for greater mobility and flexibility in the choice of places to stay, but they have also generated new inequalities and precariousness. Convenience for some people is often accompanied by employment and economic problems for others, creating an evidently complex scenario.

In the 1990s, when these companies appeared, aimed at facilitating everyday life by freeing people from material complications, for example by offering shared accommodation and managing bookings efficiently, they sought to make the experience of living in new places more accessible. In terms of transport, these platforms offered solutions that overcame the problems associated with conventional taxis, such as lack of cleanliness and comfort. They facilitated urban mobility and provided friendlier and more efficient transport options. In the area of food, they committed to providing fast and varied food options, simplifying the process of obtaining food and removing obstacles related to meal preparation and delivery.

These companies, which call themselves facilitators of freedom and labour flexibility, have created a new paradigm in which the connection between supply and demand for services is made through digital interfaces. Uber or Airbnb, among others, present themselves as catalysts of opportunity, removing material barriers and offering access to a variety of services in a seemingly free manner. They promise cheap and convenient solutions for everyday needs, from transport and accommodation to food delivery. However, behind this apparent liberation from material obstacles lies a complex web of conditions that raise fundamental ethical questions:

1.  The first condition imposed by these platforms is the absolute dependence on technology, specifically the internet. This shift towards a digital interface excludes the possibility of direct human interaction, relegating the connection between individuals to a machine–person relationship. This new dynamic redefines the way we communicate and establish working relationships.
2.  Secondly, full trust is required in the management of personal data. The significant amount of information provided to these platforms, from account numbers to personal preferences, raises concerns about how these data will be used and whether they will be handled ethically.
3.  The third and most crucial condition is the need to disengage emotionally from the people performing the services. We are asked to evaluate the system as a whole but ignore the real people behind each transaction. This precarious relationship generates fleeting interactions where the evaluation focuses more on the quality of the service than on the person providing it.

This "precariat" [32], as some call it, is not perceived as a means to build a future but as a way to earn income quickly, especially in occupations such as food delivery. This shift in the nature of work has contributed to the creation of an unequal world, where our everyday consumption choices are intrinsically linked to exploitation and precarious labour. The lack of a common narrative that encompasses our collective experiences has led to the loss of a sense of community and the difficulty of building a socially just future.

In order to take advantage of these platforms, people had to fulfil three essential conditions. First, they had to use the internet as the interface between the machine and the person. Secondly, they had to trust that the data provided would be used ethically and securely. And third, they had to bypass the personal connection with the service providers, dealing with the company rather than dealing directly with the people providing

those services. Thus, they became efficient intermediaries for overcoming material obstacles and facilitating various day-to-day activities but also posed challenges in terms of privacy, trust and personal connection. The work context that emerges from this dynamic generates an illusion of freedom for the people performing these tasks who are as free as those using the services, and, in the educational context, this illusion is further projected by building unrealistic bridges to knowledge, peers and educators, as human relationships are erased and liquid, ephemeral, unsound links are built.

The relationships implied by these platforms are characterised by their transience and the absence of a meaningful personal connection between their users. Interaction is limited to the time it takes to deliver a service, such as delivering food or driving a car or giving a "present" in a virtual classroom. In addition, it is highlighted that these people working on platforms do not see their work as a long-term career but rather as a way to earn income on a temporary basis. In this context, they are free to choose when to work or study and how much time to spend on these activities, suggesting that this flexibility translates into freedom. However, this model has also been criticised for generating precarious working conditions, with low wages, a lack of benefits and an unstable employment relationship. It is clear and concise, and it is a world that ideologically treats human relations in a way that, although it seems that we are all equal, in reality we are immersed in profoundly unequal structures [33]. The act of buying actualises, in this sense, an unequal dynamic, where in order for a product to reach your hands, there is a chain of people working in often precarious conditions to make it possible.

But what are the implications behind everyday actions, such as buying a product? Every purchase contributes to maintaining and perpetuating unequal structures, and it is essential to be aware of this in order to reflect on our role in this system [34]. From a deep perspective on the nature of our world today, we are in a position to assume that we have been dehumanising reality in many ways. Exploring the reasons behind dehumanisation and the emergence of unequal conditions is crucial to understand and address current challenges, especially to understand what (and how) our profession does. Some factors that may have contributed to this are as follows:

1. The rise of neoliberalism, which emphasises market freedom and the minimisation of state intervention, has led to increasing economic deregulation. This has resulted in the concentration of wealth in the hands of a few and more precarious working conditions for many.
2. Technology and globalisation. While technology and globalisation have brought benefits, they have also contributed to the automation of jobs, the loss of local jobs and global competition that often favours large corporations.
3. Economic inequality. the gap between rich and poor has widened, with a minority accumulating a disproportionate amount of wealth while many struggle to make ends meet.
4. A lack of awareness and action. The lack of awareness and action on the part of society at large to address these issues and demand meaningful change may also have contributed to the persistence of these conditions.

Reflecting on these issues is an important step towards understanding the challenges we face as a society and towards finding solutions that promote greater equity and humanisation in our world. The lack of a shared "grand narrative" is an interesting and complex issue. Here are some possible reasons and associated consequences:

1. Diversity of perspectives. We live in increasingly diverse societies with a multiplicity of perspectives, identities and values. Globalisation, migration and interconnectedness have brought with them a wide range of experiences and points of view. This has led to the difficulty of establishing a common narrative that resonates with all.
2. Information fragmentation. The information age has provided access to an overwhelming amount of data and opinions. However, information is often fragmented

into filter bubbles where people are exposed primarily to perspectives that already support their own beliefs, making it difficult to create a common narrative.

3. Mistrust of institutions. Mistrust of social and political institutions also plays a crucial role. Many people feel that political and economic elites do not represent them, leading to a lack of trust in a common narrative proposed by these institutions.

4. Individualism. Extreme individualism, promoted in part by neoliberal ideologies, has led to a more self-centred approach. This has weakened the idea of a common good and fostered a more atomised society.

5. Rapid technological change. Rapid technological evolution has led to social and economic change at a dizzying pace. This has left many people feeling disconnected from traditional narratives and searching for new ways to find meaning and purpose.

This lack of a common narrative can have significant consequences, such as political polarisation, social alienation and a pervasive sense of purposelessness. Bridging these cracks requires a collective effort to build bridges between different perspectives, foster empathy and seek shared values that can form the basis of a common narrative. In this context, it is important to rethink value conflicts and to engage in questions about how to ensure equity in a world marked by striking disparities. The present period, marked by significant social transformations, calls for the exploration of novel approaches to intervene in the social sphere; such an intervention must merge values with technical competencies [35].

## 5. Challenges for Professional Intervention: Implications and Recommendations to Catalyse Social Change

The above comments lead us to reflect on the processes of dehumanisation of our interactions and the loss of social consciousness [36]. The absence of common narratives has left room for a world in which individual identity is diluted in a sea of digital interfaces and impersonal transactions. Critical reflection and the search for a common narrative are essential to address these challenges and build a future where humanity and ethics are not sacrificed for the sake of convenience and digital efficiency. Everything becomes more plastic, more mouldable; this lack of a grand narrative makes everything more liquid, more gelatinous. Thirdly, the absence of narratives takes us to a strange place, where we lose the social, the notion of fighting for something social [37]. For example, these days, some groups are trying to construct a grand narrative about Israel and Gaza, but they are unlikely to succeed because of the constant manipulation of the media. It is curious how these narratives do not really construct reality and therefore do not motivate us to transform our actions. Take for example the oranges we consume, most of which come from Israel. Did it ever occur to you to say "I won't buy oranges, I'll go to the market and I won't worry if they are from Israel, Morocco, Valencia or anywhere else." Probably not. We simply buy oranges, looking for quality at the best price, without questioning their origin.

This lack of a strong narrative creates a social deficit, a problem of social awareness and an insufficient understanding that we do not all have the same opportunities. We are far from living in the right world. This, in the first place, provokes significant indignation. The imperative to reconfigure shared narratives and promote social awareness emerges as an essential component of addressing these issues and building a more just and equitable society. Social intervention proves to be a primary vehicle in this endeavour, constituting a strategic tool for modifying and shaping collective perceptions, thus bringing about significant transformations in the social structure and the promotion of fundamental values that support a more equitable order [37].

In today's society, the possibility of having a normal life seems minimal if certain aesthetic standards, such as perfect dental alignment, are not met. This raises questions about the society we live in and how certain aesthetic standards, often related to economic ability, affect our social interactions. The need to have a specific appearance, often at

considerable expense, complicates social relationships and affects the trust we place in others. This situation is aggravated by the absence of grand narratives, leading us into an ideological process where our lives are mainly made up of anecdotes. We constantly tell ourselves stories, and these anecdotes, instead of being guided by grand social narratives, become part of a personal anecdotal record. We live in a society of spectacle, where the visualisation of reality overrides all other considerations, which contributes to the creation of fragmented and narrow realities. This leads to the appearance of situations of permanent distrust towards others, as we are not clear about whether they comply with the criteria of order, hygiene, cleanliness, speak correctly or are violent or not. In the same way, strange references appear; neatness, cleanliness and order become determining factors in dealing with others. A revealing example is the fact that dental appearance, with perfectly whitened and aligned teeth, has become essential for an accepted social life.

It is reasonable to question the existence of contemporary discourse. There is, and it is the postmodern condition of which Lyotard speaks [38]. This postmodern discourse has turned us into permanent suspects, both for others and for ourselves. Suspicion has become our main narrative, and this suspicious attitude is the structural basis that gives rise to two fundamental social realities: precariousness and inequality [39]. Precarity manifests itself when we distrust everything, becoming precarious in the face of reality. Constant suspicion places us in a position of distrust towards everything around us, generating a sense of precariousness in our lives.

On the other hand, suspicion is also at the root of inequality in society. This narrative creates a social structure marked by mistrust, which in turn becomes fertile ground for the proliferation of inequalities. Although there are other problems, suspicion emerges as a key factor contributing to the formation of these complex social realities. In this portrait of a negative world, it is essential to be clear that we live in a suicidal society. We are punishing and mistreating not only ourselves but also the planet. Just as we are driving the planet to unsustainable conditions, we are also generating a socio-suicidal situation. This scenario of life on the edge, on the edge of the abyss, provokes a feeling of shame and rejection. Perversion seems to have taken root in our world, and it is difficult to convince others of this reality. Faced with this situation, two radical possibilities present themselves: to become a hero who seeks to save the planet or to resign oneself to living in indignity, becoming a kind of gravedigger.

The pessimistic and anorgasmic gaze takes us into a strange space, where every aspect of this world seems pathetic. Social workers are confronted with situations of proximity to real problems, such as domestic violence, lack of economic resources or child neglect. This proximity to stark reality poses a challenge in determining what is the minimum and maximum distance needed to address these problems. This is a crucial question in the social sciences: How do we maintain sufficient proximity to understand the problems without losing sight of the big picture? When does the distance become so great that we stop seeing the problems? Lest we approach this from a purely economistic or purely ideological perspective, which can be tempting, it is important to consider a quote from Bertolt Brecht. When fascism emerged and the human condition lost meaning, Brecht pointed out that the problem was not simply whether Jews or Nazis were good or bad but that the truth could not be told either way. Fidelity to intelligence, according to Brecht, prevented the expression of truth, as there were numerous moral tragedies, feelings of urgency, a will to experience and courage for holiness, where it was impossible to tell the truth.

## 6. Discussion and Conclusions

For those of us who work in the social sciences and seek to act on the basis of truth in order to transform it and improve the living conditions of others, this negativity places us in a peculiar place. We are faced with the choice of following the will of the market, with all its implications, or adopting a more humane and supportive approach [40]. It is a dilemma that demands reflection and conscious action. So, to return to the thread of the

argument, the question is how to intervene politically in the social sphere without falling into conventional ideological narratives and how to give importance to the people we seek to help without making us, the social workers, the centre of the narrative. It is undeniable that we cannot disengage from the market, and it is clear that we cannot ask people to become political agents of social transformation when they seek help. We must reintegrate people into their context, which is permeated by a neoliberal, market-centred ideology. The dilemma lies in how to break with biopolitics, i.e., the structuring of bodies according to a politics of fear and a hysteria of order and neatness. How do we do this? How do we confront the widespread fear of those whose teeth are not perfectly aligned, for example? This kind of questioning leads us to a profound reflection on political and social intervention in a precarious and unequal world.

In this sense, we support the idea of carrying out a social intervention that allows society to recognise the brutality of life, which would provide the opportunity to make mistakes and have new possibilities, which is the basis on which, among other things, educational transformation rests. This would mean giving back to the world the capacity to accept and learn from mistakes and, above all, to offer everyone the same conditions of accessibility to knowledge, materials, ways of working, technologies, etc. In a context where the lives of others are constantly being observed, social action has the potential to be indispensable in accounting for the difficulties and sufferings faced by some people, which is deepened by the high rates of school failure in a large part of the world. The real opportunity of such work and education lies in giving back to society the possibility of seeing itself, of recognising the unstructured and dislocated realities that exist. In doing so, it restores, in a way, the capacity to confront and address its own problems. This implies an approach that goes beyond simply solving individual problems; it is about questioning and challenging the social structures that generate these issues [41]. Ultimately, social intervention should not simply be a tool to integrate people into the market, education or any other specific social field but rather a force to question and transform the conditions that generate suffering and inequality. This approach requires moving away from neoliberal logic and focusing on building a society that allows for diversity, equity and the ability to learn from our own imperfections, and even more, to design modes of social tolerance that break with the conditions of exitism, perfection and beauty that undermine our imperfect individualities.

The task of summarising these reflections presents a challenging one, but it essentially raises the question of the existence of utopias in contemporary times. The premise that utopias no longer exist, or at least not in their classical form, suggests the possibility that we may in fact live in one where, for many, today's society may be characterised by relative equality, democracy and credibility, albeit with shades of precariousness and significant differentiation. It is argued that perceptions of utopia can vary widely between individuals, with voices denying the existence of structural problems and seeing precarity as an inevitable price in a society they see as profoundly egalitarian. However, questions arise about meritocracy in institutions such as universities, where certain criteria may exclude women, leading to a dichotomy between modifying the meritocracy model to ensure the inclusion of all women or maintaining it in pursuit of principles of equality and justice.

This dichotomy reflects the search for an alternative utopia, although authors such as Alain Badiou [42] suggest that the reality we live in is itself a utopia, constructed virtually. The virtuality of our interactions raises the possibility that the perceived utopia may vanish in a face-to-face encounter, as the virtual image may distort individual reality. In this context, the importance of socio-educational action is highlighted as a means to explore and account for underlying realities, challenging virtual constructs and embracing the courage to immerse oneself in less-favoured environments. The fundamental mission of the social professions' intervention, beyond generating reports, lies in the need to confront tangible reality and communicate the existence of real worlds, which coexist with virtual worlds.

Precarity manifests itself as the lack of opportunities that certain people experience in relation to various services and resources. This phenomenon implies that some people do not enjoy the same opportunities as others in accessing different resources. An illustrative example is the disparity in access to higher education, where some women face obstacles in accessing university. It is relevant to note that precariousness is not limited to an international context but is also observed at the national level. In this sense, there are women who do not have the opportunity to access university education, which underlines the inequalities that exist even within the same country. In a broader sense, precariousness encompasses difficult circumstances in which people are unable to meet their basic needs, such as food and shelter. These difficulties arise as a result of a lack of opportunities throughout their lives.

Precariousness, from another point of view, is defined as the lack of resources to satisfy basic needs, highlighting its character as a necessity rather than a whim. Ana María Romero [43] adds that precariousness not only implies a lack of needs but also contributes significantly to inequality, alluding to circumstances and luck in life that determine access to certain resources. Some theorists broaden this perspective and consider it as the condition of possibility in which people may find themselves in various spheres. The discussion reveals that precarity is not only related to a lack of resources but also reflects the unequal distribution of these resources, depending on factors such as place of birth or life circumstances. The reflections of some critics, such as Standing [32], highlight the importance of thinking beyond the individual definition of precariousness and advocate for a community perspective. The idea is put forward that a lack of resources does not necessarily indicate a lack of resources but rather an unequal concentration in the hands of a few [44]. It is illustrated with the example of the collective purchase of solar panels, underlining how the lack of access to information and the lack of collective thinking contribute to the perpetuation of precariousness. Personal testimony about the purchase of solar panels highlights how a lack of resources is also linked to a lack of collective awareness and organisation. The anecdote highlights the need to rethink the way in which issues of precarity are addressed and suggests that the lack of resources may be more a question of distribution and access than their actual non-existence.

More than a few thinkers conclude the discussion by highlighting the fundamental role of community in tackling precarity. The problem of the lack of a collective idea of community is raised as a serious obstacle in the fight against precarity and other social problems. This fundamental questioning underlines the need to rethink and reconstruct fundamental concepts of solidarity and community in today's society. This reflection highlights a crucial point about the individual and collective perception of precariousness: there is a tendency to see precariousness as a purely individual problem, under the belief that each person can solve his or her challenges independently. However, there are circumstances in which collective awareness and community action are essential to address and find solutions to problems related to precarity. This perspective highlights the idea that precariousness is not simply an individual problem but a social challenge that requires collective efforts to address it effectively. Training in the socio-educational field is also mentioned, underlining that from the beginning of this training, the importance of understanding that problems such as mental health, a lack of employment or the inability to pay for basic services are not individual anecdotes but serious and systemic problems that affect society as a whole is emphasised.

Finally, a critical perspective on the idea of work in today's society is raised, suggesting that the problem lies not in the lack of work but in the abundance of work. This observation points to deeper questions related to the nature of work, its distribution and the balance between work and other aspects of life. This critical approach invites a rethinking of entrenched concepts about work in contemporary society. For example, work overload and the lack of work–life balance, as well as the critique of the work system that imposes long working hours and affects the quality of life to such an extent that it becomes a further edge of an enslaving system. Moreover, the observation about motherhood and

gender inequality is particularly relevant; the feminist struggle should not only focus on equality in the workplace, as this is an environment that can be demeaning and abusive even for men. Feminist struggles continue to make a break to include equality in other aspects of life, such as parenting and childcare, recognising the differences in bodily wear and tear and the responsibilities associated with childbearing or education at different levels of children's lives. The call to politicise people's consciousness through socio-educational intervention is indisputable in such a way that helping to find employment is not the sole purpose, but rather it is about providing subjects with the tools for a minimum political understanding, fostering awareness of the impact of work on society and its implications beyond the individual sphere. This suggestion highlights the importance of addressing not only people's immediate challenges but also promoting a critical and conscious understanding of their role in diverse environments and human groups. There are no recipes for disarticulating the logics put forward; on the contrary, there remain topos, common places where to look for questions and possible answers that lend themselves to the game of denaturalising practices that attempt to erase all traces of subjectivity in our labour and educational relations, relations that are, first and foremost, human.

**Author Contributions:** Conceptualization, J.-L. A.-F., and R.G. P. G.; methodology, J.-L.A.-F.; validation, R.G., M.d.C.S.-M. and R.G.P.G., ; formal analysis, M.d.C.S.-M.; investigation, J.-L.A.-F..; resources, R.G.P.G.; data curation, R.G. ; writing—original draft preparation, J.-L.A.-F.; writing—review and editing, R.G.., M.d.C.S.-M. and R.G.P.G.; supervision, M.d.C.S.-M. and R.G.P.G.; project administration, J.-L.A.-F. and M.d.C.S.-M.; funding acquisition, J.-L.A.-F. and M.d.C.S.-M.

All authors have read and agreed to the published version of the manuscript.

**Funding:** This research received no external funding

**Data Availability Statement:** No new data were created or analyzed in this study. Data sharing is not applicable to this article.

**Conflicts of Interest:** The authors declare no conflict of interest.

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
