# Peer review of "Critical Reflection on Inequality and Precariousness in Contemporary Society: Reconstructing Connections from Social Intervention"

_education, doi:10.3390/educsci14040379_

Round 1

Reviewer 1 Report

Comments and Suggestions for Authors

Very insightful article which developed critical thinking and societal frameworks within socio-education lens. Introduction was well aligned with key concepts and flowed well.

Abstract is informative, may add last lines a future suggestions or inform reader about conclusion drawn from your research. 

Demonstration of excellent points specifically as drawn within lines#30-36.

Historical context gives reader a desired background regarding societal issues faces inequalities #40-53

#138 ( Section 3) may suggest a sub heading to organize important literature prejudices, historical workplace environments etc....

Great descriptions within remaining segments that align with topic and flow together. 

#580-598 A very powerful conclusion that gives explicit examples of your work and research . You may add specific details as to how we can impact these changes, what exactly can we as society do to remedy these enslavement practices at workplace? 

Very interesting and current issues within society...

References are well aligned and developed 

Author Response

Responses to reviewers: 
reviewer 1: 
- the division into subheadings of section 3 was taken into account; this resulted in organizing the contributions into two dimensions (gender and machinization of the human).
- points 5 and the conclusion were taken up in order to suggest some actions to favor social intervention.

Reviewer 2 Report

Comments and Suggestions for Authors

I recommend that the article undergo some reconsiderations before publication. My rationale for this recommendation is that the presented research exhibits very good potential but requires a better structure and more specificity. The following suggestions are offered for the consideration of the authors:

The introduction introduces two primary concepts for analysis: inequality and precariousness. However, it fails to clearly delineate the context of analysis: whether it pertains to a social, societal, or educational framework. Furthermore, the connection between these analytical units and socio-educational intervention, the second variable presented in the title, remains ambiguous. In a critical analysis article, the introduction should establish the context and introduce the topic of reflection, providing pertinent background information and theoretical frameworks. Therefore, it is imperative to clarify all variables, their interconnections, and the theoretical frameworks guiding the analysis.

The second part of the article, "A Socio-Historical Look at Social Inequalities," effectively outlines the evolution of social inequity until the 19th century. However, there is a lack of clarity regarding the transition to the 21th century and currect implications.

The stated purpose from the abstract of the article is `to build a solid conceptual framework that allows for the re-evaluation of interventions in this complex environment, critically approaching the thought of new modalities of action`. It seems to me that this goal was not achieved, a clear conceptual framework was not presented, the social interventions were not clearly reviewed, they were only discussed in the discussion section. I did not notice a clear connection between the analyzed concepts: inequality precariousness and social interventions. The stated goal can be reanalyzed.

Although you can see a vast knowledge of the issue and a very good power of analysis, the speech seems poorly structured and difficult to follow. Therefore, I recommend a restructuring to improve understanding: specific for a critical analysis article are the following points: 1.the introduction of concepts and the presentation of the current state in the research of these concepts; 2. The critical analysis of concepts, which involves questioning assumptions, challenging dominant narratives, exploring contradictions, and considering multiple perspectives; 3. personal reflections, on the beliefs, values, and biases in relation to the topic; 4. the theoretical framework: drawing on relevant theoretical perspectives; 5. implications for practice and research and recommendations for social change.

Additionally, considering the journal's focus on education, there appears to be a limited exploration of the intersection between education and inequity within the societal context. I recommend incorporating more focused reflections on the educational landscape to enrich the analysis and deepen our understanding of how inequities manifest and perpetuate within educational systems.

Sincerely,

Author Response

- the objective was reformulated: what was really done was not to "construct a conceptual framework" but to critically reflect on some dimensions of a broad conceptual framework of a social and anthropological nature.
- in the introduction, the background belonging to historical materialism and the dialectics of social processes are more clearly recognized
- recommendations and suggestions for social transformation were detailed towards the end of the article (item 5 and conclusions)
- small clarifications were introduced in several sectors of the text, showing that when we speak of educational and labor fields where inequalities and precariousness are expressed. 

Translated with DeepL.com (free version)

Round 2

Reviewer 2 Report

Comments and Suggestions for Authors

Congratulations on your research! Great effort!

I revised this article (which proposes a critical reflection on inequalities in contemporary society) in its first and second rounds. 

I analyzed the integration of the suggestions I made in the first round of reviews, the authors' response and the final quality of the article.

Following my analysis, I propose publishing the article in its current form, without further review.

The reasons for my decision are the following:

The authors respected the suggestions made: The purpose of the study was reformulated, to express more precisely the nature of the research. Clarifications were made on the type of intervention analyzed. The structure of the speech has been improved, by introducing sub-chapters for a clearer structure. The authors have introduced references to education throughout the article.

The current quality of the article is good, with a good level of English.

Congrats!